# Effect of Prolonged Serum Storage Time and Varied Temperatures on Biochemical Values in African Savanna Elephants (*Loxodonta africana*)

Emily L. Schlake [1], Katherine R. Cassady [1], Erika J. Gruber [2] and Larry J. Minter [1,3,*]

1 Department of Clinical Sciences, North Carolina State University, Raleigh, NC 27606, USA
2 Department of Population Health and Pathobiology, North Carolina State University, Raleigh, NC 27606, USA
3 North Carolina Zoo, 4401 Zoo Pkwy, Asheboro, NC 27205, USA
* Correspondence: jb.minter@nczoo.org

**Abstract:** Blood samples are routinely collected from wild populations in remote locations with limited electricity, minimal diagnostic capabilities, and extreme environmental conditions. Under these conditions, serum samples may be stored for prolonged time under varied temperatures prior to processing, which could affect the ability to interpretation the results. This study's objective was to evaluate the effects of delayed processing of serum samples and varied storage temperatures on biochemical values in African savanna elephants (*Loxodonta africana*). Blood samples were collected from six elephants managed by the North Carolina Zoo. For each elephant, seven red top tubes were collected. One serum sample for each elephant was analyzed on Day 0 (control group). The remaining samples were stored under different temperatures including room temperature (23 °C), refrigeration (2.2 °C), and incubation (32.2 °C), with samples from each temperature group being analyzed on Day 5 and Day 10. Many of analytes (10 out of 18) did not change significantly regardless of storage temperature or time. Refrigeration improved stability in an additional four analytes over prolonged storage. We conclude that if serum is properly separated shortly after collection, many serum biochemical analytes can be accurately measured even after suboptimal serum storage, but refrigeration and prompt evaluation are still required for some analytes.

**Keywords:** elephant; serum chemistry; blood storage; temperature; serum stability; *Loxodonta africana*

## 1. Introduction

Serum biochemistry analysis provides a lot of information about the health of individuals and populations of animals. For the most accurate measurements, many diagnostic laboratories recommend that serum be separated from cells as soon as clot formation occurs and refrigerated at 4 °C if it cannot be analyzed immediately [1–3]. However, wildlife field researchers working in remote field sites often do not have access to the equipment needed to preserve samples for future analysis. Wildlife researchers and veterinarians collect blood from animals in the wild to compare to populations in human care [4–6], to appreciate geographical differences [7,8], and to monitor the health status of wild populations [9,10]. The blood samples collected are used to investigate the genetic diversity, hormone concentrations, reproductive status, and more for populations [11,12]. Often, researchers in the field are working far away from diagnostic labs, reliable electricity, and refrigeration systems; so biological samples may not be refrigerated prior to processing. Portable blood analyzers are available, but they are expensive, may be nonfunctional without electricity and access to refrigeration for test cartridges [13]. For wildlife researchers, this means that they must find the best way to handle and store samples using the means available to them, while ensuring the most accurate analysis of their samples.

The effect of delayed time to serum separation on specific biochemical analytes has been evaluated. Delayed separation can result in decreased glucose, increased phosphorus,

and elevated lactate [1,2,14–17]. This suggests timely separation of serum from cells while in the field is worth the effort and investment into field centrifuges [11,18–20]. Assuming that these biological samples could be centrifuged in the field with a portable non-electric centrifuge [18,21], the effects of prolonged storage before analysis needs to be evaluated. Current literature evaluating the effect of sample storage temperatures and delayed centrifugation effects is focused on humans and domestic species. These studies found more biochemical analyte stability in samples that were processed with limited storage time and stored under refrigeration (4 °C) or freezing (−20 °C) temperatures [3,22–26]. Nondomestic species studies have focused on the storage of whole blood samples and the effect of delayed centrifugation prior to analysis [2,15], but the effect of delayed processing of biochemical values has not been evaluated in any elephant species.

African savanna elephants (*Loxodonta africana*) are currently listed as endangered and African forest elephants (*Loxodonta cyclotis*) are currently included as critically endangered on the International Union for Conservation (IUCN) Red List of Threatened Species [27,28]. Conservation efforts rely on information obtained through field research, but both elephant species live in remote locations, which means that delayed sample processing is unavoidable. The effect of prolonged storage on serum samples needs to be better understood for accurate interpretation of biochemical findings from these animals.

The objective of this study was to determine the effects of storage temperature and time on measurement of serum biochemical analytes in African savanna elephants. We hypothesized that increased storage temperature and time would decrease the reliability of serum biochemistry testing.

## 2. Materials and Methods

Subjects included six African savanna elephants (*Loxodonta africana*), two males and four females. Animals were housed at the North Carolina Zoo and were trained to allow for voluntary blood collection. This study was reviewed and approved by the North Carolina Zoo research review committee, which acts in lieu of an Institutional Animal Care and Use Committee (IACUC). Blood samples were collected from the posterior auricular vein using a 21-gauge butterfly catheter with a vacutainer attachment (Becton, Dickinson, and Company, NJ 07417, USA) to allow for immediate filling into seven separate red top tubes for each elephant. The blood samples were allowed to clot in a container without ice while transported to the on-site veterinary hospital and centrifuged within an hour of collection. Samples were centrifuged at $4000 \times g$ for 10 min within one hour of collection in a Themo Scientific Heraeus Clinifuge centrifuge (Waltham, MA, USA). Serum was transferred to a Fisherbrand™ cryogenic storage tubes (Waltham, MA, USA) after centrifugation. Biochemical analyses were performed on a Beckman Coulter AU5800 analyzer (Brea, CA, USA) at a veterinary diagnostic laboratory (Antech Diagnostic Laboratory; Chapel Hill, NC, USA) approximately 60 miles from the North Carolina Zoo. Biochemical analysis was performed for each animal on the day of blood collection to serve as the control. The remaining six serum aliquots per elephant were stored at the North Carolina Zoo Veterinary Hospital. Serum samples were stored under the recommended storage condition (refrigeration, 2.2 °C), and at two temperatures to reflect potential field conditions: room temperature (approximately 23 °C) and heat (32.2 °C). Heated samples were stored in a temperature-controlled incubator (Animal Intensive Care Unit, Animal Product, Noroc, CA, USA). On Days 5 and 10 post-collection, biochemical analysis was performed on serum stored at each of the three temperatures. Samples were refrigerated during vehicle transport to the diagnostic laboratory.

Statistical analysis was performed using the JMP software (JMP, version 16.0, SAS Institute, Cary, NC, USA) to determine the effect of delayed sample processing and storage temperature on biochemistry values. All of the variables were tested for normality of distribution by the Shapiro–Wilk test. Significant differences between Day 0 samples for each biochemical analyte and each treatment group were assessed using Wilcoxon signed rank tests with Bonferroni correction applied. The results are reported as mean ± standard

deviation (SD) and were considered to be significant at $p \leq 0.05$. The means of each test group were compared to Day 0 results (control group) and statistically significant differences were identified. The values were then considered in terms of percentage of change compared to control values from Day 0. Graphs were prepared in GraphPad Prism [Prism, version 9.4.1, GraphPad Software, Inc., San Diego, CA, USA).

## 3. Results

No statistically significant differences were detected for the measurement of total protein, globulin, albumin, gamma-glutamyl transferase (GGT), creatinine, phosphorus, glucose, sodium, potassium, chloride, or cholesterol at any temperature or time. In contrast, we observed time and temperature-dependent decreases in aspartate aminotransferase (AST) and creatinine kinase (CCK) activity. Lactate dehydrogenase (LDH) activity was decreased in refrigerated samples evaluated on Day 5 and 10, but somewhat unexpectedly storage at higher temperature appeared to mitigate this effect. Alkaline phosphatase (ALP) activity was relatively stable, with the only decrease in ALP observed in the samples stored for 10 days in heat. Compared to Day 0, blood urea nitrogen (BUN) was decreased on Day 5 and Day 10, but temperature did not have any additional effect. Minor, but statistically significant, decreases in total bilirubin were noted in samples stored at room temperature and heat for both 5 and 10 days. Full results are summarized in Table 1. For each analyte with a statistically significant changes with storage, the percent change was calculated and reported in Table 2.

**Table 1.** Biochemical values of African savanna elephants (Loxodonta africana) under different storage lengths and temperatures.

| Day of Analysis | Day 0 (Control) | Day 5 | | | Day 10 | | |
|---|---|---|---|---|---|---|---|
| Storage Temp | Not Applicable | Refrigeration (2.2 °C) | Room Temp (~23 °C) | Heat (32.2 °C) | Refrigeration (2.2 °C) | Room Temp (~23 °C) | Heat (32.2°C) |
| Total Protein (g/dL) | 7.55 ± 0.54 | 7.51 ± 0.41 | 7.48 ± 0.39 | 7.50 ± 0.31 | 7.38 ± 0.39 | 7.35 ±0.42 | 7.33 ± 0.34 |
| Albumin (g/dL) | 2.90 ± 0.14 | 2.88 ± 0.04 | 2.88 ± 0.07 | 2.83 ± 0.10 | 2.86 ± 0.10 | 2.81 ± 0.08 | 2.83 ± 0.12 |
| Globulin (g/dL) | 4.65 ± 0.61 | 4.55 ± 0.57 | 4.50 ± 0.54 | 4.60 ± 0.55 | 4.48 ± 0.54 | 4.5 ±0.55 | 4.45 ± 0.51 |
| AST (IU/L) | 17.1 ± 2.31 | 16.1 ± 2.14 | 14.3 ± 2.16 * | 12.0 ± 1.41 * | 16.3 ± 2.16 | 12.0 ± 1.55 * | 8.5 ± 1.04 |
| ALP (IU/L) | 60.2 ± 5.91 | 57.3 ± 6.56 | 57.0 ± 6.06 | 57.0 ± 7.46 | 56.1 ± 7.94 | 54.0 ± 4.98 | 52.5 ± 6.22 * |
| GGT (IU/L) | 8.0 ± 0.89 | 8.0 ± 0.89 | 7.6 ± 0.82 | 8.0 ± 0.89 | 8.0 ± 0.89 | 7.6 ±0.82 | 7.8 ± 0.98 |
| Total Bilirubin (mg/dL) | 0.16 ± 0.05 | 0.16 ± 0.05 | 0.10 ±0.0 * | 0.10 ± 0.0 * | 0.16 ± 0.05 | 0.10 ± 0.0 | 0.10 ± 0.0 * |
| BUN (mg/dL) | 11.1 ± 1.94 | 8.8 ± 0.98 * | 9.0 ± 1.10 * | 9.0 ± 1.09 * | 9.0 ± 1.55 * | 8.83 ± 0.41 * | 9.16 ± 1.33 * |
| Creatinine (mg/dL) | 1.30 ± 0.25 | 1.16 ± 0.26 | 1.23 ± 0.23 | 1.18 ± 0.21 | 1.23 ± 0.25 | 1.25 ± 0.23 | 1.23 ± 0.22 |
| Phosphorus (mg/dL) | 4.10 ± 0.18 | 4.15 ± 0.15 | 4.15 ± 0.21 | 4.18 ± 0.11 | 4.46 ± 0.37 | 4.25 ± 0.40 | 4.30 ±0.36 |
| Glucose (mg/dL) | 4.10 ± 0.18 | 81.1 ± 6.24 | 81.66 ± 6.53 | 74.0 ± 5.75 | 82.5 ± 9.01 | 79.5 ± 9.81 | 80.6 ± 10.38 |
| Calcium (mg/dL) | 10.10 ± 0.34 | 10.10 ± 0.29 | 10.08 ± 0.16 | 10.16 ± 0.19 | 10.00 ± 0.12 | 9.68 ± 0.70 * | 9.8 ± 0.55 |
| Sodium (mEq/L) | 128.0 ± 1.17 | 129.1 ± 1.16 | 129.3 ± 0.81 | 129.0 ± 1.09 | 130.3 ± 1.63 | 130.8 ±2.40 | 130.5 ± 2.07 |
| Potassium (mEq/L) | 4.50 ± 0.23 | 4.53 ± 0.19 | 4.53 ± 0.16 | 4.56 ± 0.21 | 4.51 ± 0.20 | 4.58 ± 0.17 | 4.58 ± 0.20 |
| Chloride (mEq/L) | 88.1 ± 0.75 | 88.1 ± 0.41 | 88.3 ± 0.51 | 88.3 ± 0.52 | 87.6 ± 0.52 | 88.1 ± 0.75 | 88.5 ± 1.97 |
| Cholesterol (mg/dL) | 79.5 ± 18.5 | 77.3 ± 17.06 | 77.83 ± 16.94 | 78.0 ± 17.17 | 74.3 ± 15.88 | 74.6 ± 18.28 | 73.6 ± 15.24 |
| Creatinine Kinase (IU/L) | 228.6 ± 36.61 | 213.6 ± 32.82 | 185.8 ± 40.89 | 146.8 ± 28.82 * | 205.0 ± 43.50 | 174.5 ± 35.84 * | 133.16 ± 39.30 * |
| LDH (IU/L) | 308.5 ± 48.45 | 192.3 ± 53.32 * | 275.8 ± 51.15 | 271.3 ± 50.32 | 154.5 ± 40.91 * | 232.6 ± 66.49 * | 230.6 ± 62.87 * |

Results are reported as statistical mean of each analyte followed by standard deviation. * Results that are significantly different compared to Day 0 values (*p*-value < 0.05) are marked with an asterisk (*).

**Table 2.** Percent change of statistically significant biochemical analytes from African savanna elephants (*Loxodonta africana*) after serum storage under refrigeration (2.2 °C), room temperature ~23 °C), and heat (32.2 °C) compared to Day 0 values.

| Analyte | Time Comparison (Days Post-Collection) | Storage Temperature Comparison | Average Change |
|---|---|---|---|
| AST | Day 0 vs. Day 5 | Room Temperature | 16.4% decrease |
| | Day 0 vs. Day 5 | Heat | 29.8% decrease |
| | Day 0 vs. Day 10 | Room Temperature | 29.8% decrease |
| | Day 0 vs. Day 10 | Heat | 50.3% decrease |
| ALP | Day 0 vs. Day 10 | Heat | 13.0% decrease |
| Total Bilirubin | Day 0 vs. Day 5 | Room Temperature | 37.5% decrease |
| | Day 0 vs. Day 5 | Heat | 37.5% decrease |
| | Day 0 vs. Day 10 | Room Temperature | 37.5% decrease |
| | Day 0 vs. Day 10 | Heat | 37.5% decrease |
| BUN | Day 0 vs. Day 5 | Refrigerated | 20.7% decrease |
| | Day 0 vs. Day 5 | Room Temperature | 18.9% decrease |
| | Day 0 vs. Day 5 | Heat | 18.9% decrease |
| | Day 0 vs. Day 10 | Refrigerated | 18.9% decrease |
| | Day 0 vs. Day 10 | Room Temperature | 20.5% decrease |
| | Day 0 vs. Day 10 | Heat | 17.5% decrease |
| Phosphorus | Day 0 vs. Day 10 | Refrigerated | 8.8% increase |
| Calcium | Day 0 vs. Day 10 | Room Temperature | 4.2% decrease |
| Creatine Kinase | Day 0 vs. Day 5 | Heat | 35.8% decrease |
| | Day 0 vs. Day 10 | Room Temperature | 23.7% decrease |
| | Day 0 vs. Day 10 | Heat | 41.7% decrease |
| LDH | Day 0 vs. Day 5 | Refrigerated | 37.7% decrease |
| | Day 0 vs. Day 10 | Refrigerated | 49.9% decrease |
| | Day 0 vs. Day 10 | Room Temperature | 24.6% decrease |
| | Day 0 vs. Day 10 | Heat | 25.3% decrease |

Refrigeration of samples improved the stability of most of the remaining analytes, with 14 of the 18 total analytes remaining statistically unchanged for the prolonged storage time when refrigerated after collection. Along with the 10 analytes that were unchanged under all storage conditions, AST, ALP, calcium, and creatinine kinase had improved stability and remained statistically unchanged through ten days of storage when refrigerated at 2.2 °C.

Blood urea nitrogen and LDH were statistically decreased at both Day 5 and 10 of storage when refrigerated. BUN showed statistically significant decreases under all storage conditions and storage times. LDH in the samples was unexpectedly not significantly different at Day 5 under room temperature and incubation but was significantly decreased under all other storage conditions.

Serum samples stored at room temperature or under heated conditions did have significant changes for several analytes, including AST, ALP, total bilirubin, and creatinine kinase. Aspartate aminotransferase was significantly decreased when stored at room temperature and when incubated at 32.2 °C but was normal with prolonged storage if refrigerated. Alkaline phosphatase was only significantly decreased with prolonged storage under incubation but was within a statistically acceptable range under the other experimental conditions. Total bilirubin was most stable when refrigerated but showed statistically significant decreases when stored at room temperature and under incubation. Total bilirubin was decreased consistently by 37.5% to 0.10 mg/dL (SD = 0.0) when stored at both ~23 °C and 32.2 °C. Creatine kinase was statistically unchanged when stored under refrigeration but was significantly decreased when incubated for all time periods or stored at room temperature for ten days.

## 4. Discussion

To ensure validity of biochemistry results, recommendations indicate serum should be stored at 4 °C if it cannot be analyzed immediately. However, optimal storage conditions are often not practical for wildlife fieldwork, and biological samples may be stored at warmer temperatures ranging from room temperature (around 23 °C) to much warmer environmental temperatures (e.g., 32 °C). Our results indicate that many common biochemical analytes are stable for up to ten days at temperatures up to 32.2 °C.

Reliable serum biochemistry results require prompt centrifugation and serum separation shortly after blood collection [1,2,14–17]. Non-electric field centrifuges have been evaluated to allow for timely serum separation without need for a power source, which addresses one hurdle in field medicine [11,18]. If serum samples cannot be analyzed immediately, refrigeration at 4 °C is recommended to minimize storage-related artifacts [1–3]. Several studies in domestic species have evaluated the effect of storage at room temperature and refrigeration on routine biochemical analytes, but none have evaluated elephant samples [3,22–26]. More broadly, to our knowledge, no studies have investigated the effect of storage in temperatures above room temperature, an important consideration in field studies conducted in hot climates.

Our study suggests that many biochemical analytes are stable in elephant serum stored for up to ten days of storage at temperatures up to 32.2 °C. Analytes with high stability included total protein, globulin, albumin, GGT, creatinine, phosphorus, glucose, sodium, potassium, chloride, and cholesterol. Stability of the inorganic analytes (e.g., sodium, potassium, chloride, and phosphorus,) was expected since they do not degrade and because the potential cellular components that could alter these concentrations are absent in serum [29]. Glucose was also stable in stored elephant serum, which corroborated previous studies that demonstrate that serum glucose is stable if serum is promptly separated from the cellular blood constituents that continue to utilize glucose for cell metabolism [1,2,15–17]. Cholesterol has previously been shown to remain stable despite storage temperature [30]. Previous studies have shown that total protein, albumin, and globulin are relatively stable in temperatures up to 4 °C [2,16,17,23]. Since many proteins are heat labile, it was somewhat unexpected that we observed that total protein, albumin, and globulin were stable in the conditions tested in the current study. Although not reported to be stable in previous literature, GGT and creatinine also remained statistically unchanged despite our varied study conditions. We conclude that these analytes, which remained stable despite their storage conditions, can be reliably measured in serum samples stored between 2.2 °C to 32.2 °C for up to ten days.

In contrast to the above analytes, several analytes showed significant change under our experimental storage conditions as listed in Table 2 and graphed in Figure 1. AST, ALP, and CK were particularly sensitive to storage at 32.2 °C. Measured enzymatic activity for each of these analytes was lower in serum stored at to 32.2 °C compared to 23 °C or 2.2 °C as visualized in Figure 1. Alterations in these values can significantly impact clinical interpretation of an individual's health if the samples have been stored under heated conditions and may lead a clinician to miss evidence of liver disease or muscle injury. In alignment with best practice recommendations, we conclude that measurement of these analytes should be performed immediately, if possible. When immediate analysis is impractical, serum should be stored at 2.2 °C. Storage at −20 °C may further improve stability of these analytes [31,32]; however we did not evaluate this condition as this is not a practical storage condition for field research.

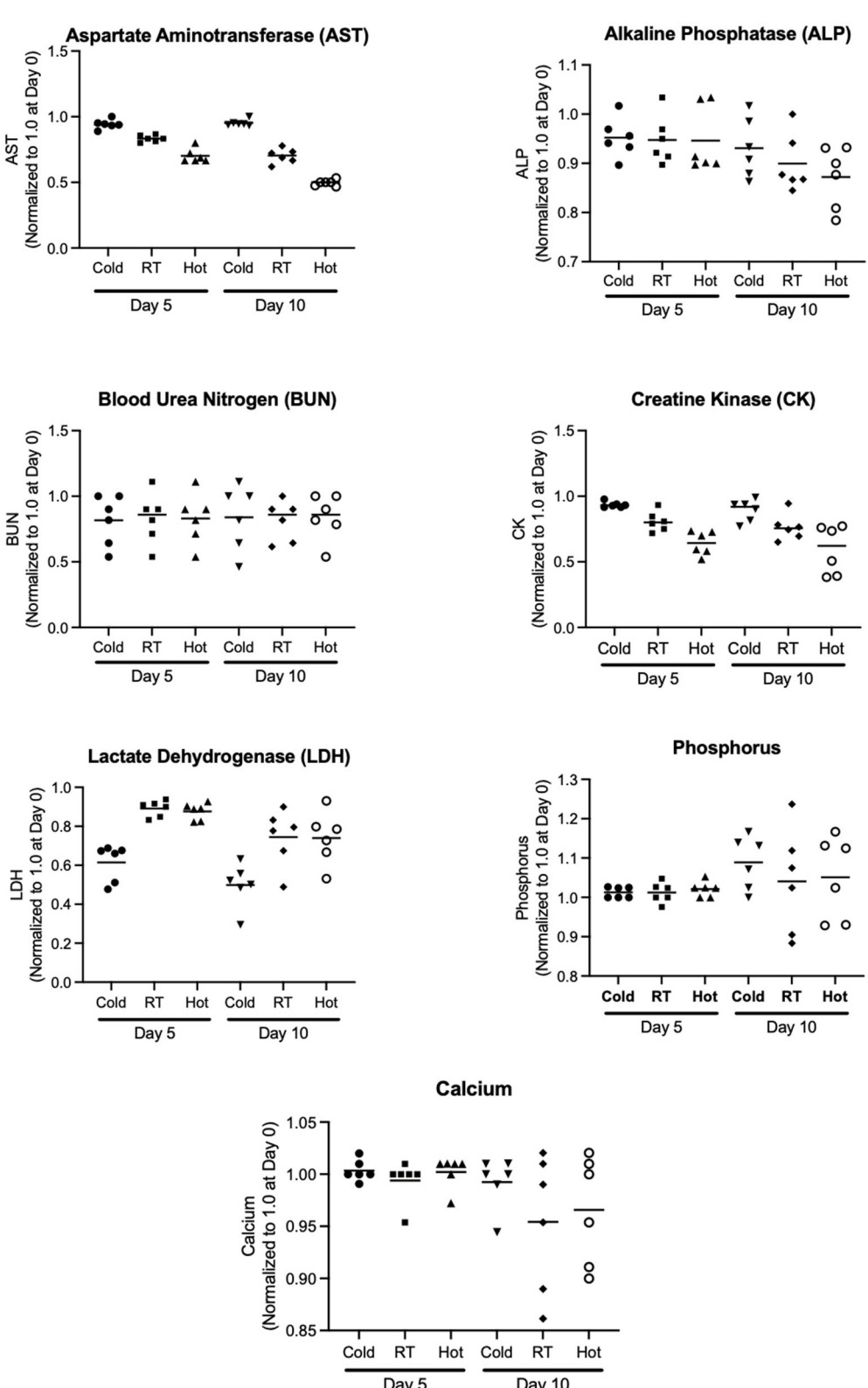

**Figure 1.** Serum biochemical analytes (AST, ALP, BUN, CK, LDH, phosphorus, and calcium) of African savanna elephants with significant changes (*p*-value ≤ 0.05) when stored at 2.2 °C ("Cold"), ~23 °C (Room Temperature, "RT"), or 32.2 °C ("Hot") and then analyzed at Day 5 and Day 10 post-collection. Data were normalized to Day 0 (control group) by converting the biochemical values to percentages using Day 0 values designated as 1.0. Each data point represents a result from one of the six elephants for each storage condition. Each test condition is represented by a different data point shape to allow easy differentiation of test groups in the graphs. The linear bars represent the mean.

Significant variability in two analytes, BUN and LDH, were observed with prolonged storage, regardless of the temperature (Figure 1). The reported values of BUN, in particular, were significantly decreased under all storage conditions. While the values were considered significantly altered, the values were still within the published reference intervals for African elephants [33]. In individuals without other evidence of kidney or liver disease, a low BUN may not be considered clinically important, so this alteration may not be clinically significant. However, in individuals with marked kidney disease, falsely decreased BUN could lead to clinicians missing this indicator of kidney disease. Our study suggests that BUN can only be accurately interpreted when analyzing fresh samples or samples stored for only short periods of time.

Our study found three analytes which did not follow a specific pattern of temperature- or time-dependent change, including LDH, phosphorus, and calcium. LDH was decreased under 2.2 °C at Day 5 and was significantly decreased under all storage conditions at Day 10. While laboratory error is a possible consideration, these alterations cannot be easily explained. With both phosphorus and calcium, only a single storage condition resulted in significantly different values. However, the values were more variable at Day 10 compared to the Day 0 or Day 5 samples as shown in Figure 1. Although a statistically significant difference in phosphorus was identified in samples stored at 2.2 °C after ten days, no statistical differences were observed at other temperatures. We speculate that this is likely a spurious result. The reported difference of phosphorus at 2.2 °C at Day 10 was mild, suggesting that it is unlikely to be clinically relevant if the variability is a true difference. The statistically significant difference in calcium was reported at 23 °C at Day 10, but there were no differences reported at any other temperature. This change was likely driven by our limited sample size and a random error in one of the samples, rather than a true phenomenon.

Our study has several important limitations. First, we tested only a small sample size of healthy captive adults, resulting in a narrow range of serum biochemical analytes. It is possible that our study is insufficiently powered to identify more subtle differences in analyte concentrations. Additionally, it is possible that storage effects are greater when serum from diseased or wild animals is evaluated. The analytes studied were also fairly limited and did not include cytokines, acute phase proteins, hormones, or other more analytes that may be more heat- or time-labile. Additional consideration of more frequent testing intervals (e.g., daily testing rather than every 5 days) should also be considered to better understand the minor changes occurring between the measured time periods. While beyond the scope of the current investigation, development and validation of correction formulae to estimate original analyte concentrations from concentrations measured in serum stored at different temperatures for different lengths of time could provide field researchers and veterinarians with an additional tool. Lastly, serum was separated using an optimized laboratory centrifugation. To truly replicate field studies, in which laboratory centrifugation is not available, serum separated by a suboptimal centrifugation technique could be evaluated in combination with storage temperatures. Future studies should include a larger, more demographically diverse population and consider additional variables like centrifugation techniques and additional analytes. It is unknown whether serum from other species would have similar analyte stability as observed in the elephant samples here.

In conclusion, shorter storage time resulted in more consistent biochemical results across nearly all reported analytes, particularly when samples were stored under refrigeration. Whenever possible, it is recommended to analyze samples as soon as possible to avoid storage-related changes of analytes. However, many serum biochemical analytes can be interpreted even after prolonged storage under temperatures up to 32.2 °C. This opens up new opportunities for blood analysis interpretation despite less-than-ideal (suboptimal sample processing in) field conditions.

**Author Contributions:** Conceptualization, K.R.C., E.J.G. and L.J.M.; methodology, K.R.C., E.J.G. and L.J.M.; formal analysis, E.L.S., L.J.M.; investigation, E.L.S., K.R.C., E.J.G. and L.J.M.; resources, L.J.M.; data curation, E.L.S. and L.J.M.; writing—original draft preparation, E.L.S.; writing—review and editing, E.L.S., L.J.M., K.R.C. and E.J.G.; visualization, E.L.S.; supervision, K.R.C., E.J.G. and L.J.M.; project administration, L.J.M.; funding acquisition, L.J.M. All authors have read and agreed to the published version of the manuscript.

**Funding:** This research was funded by the generous support of the North Carolina Zoo's Hanes Veterinary Hospital.

**Institutional Review Board Statement:** The work in this study was approved by the North Carolina Zoo research review board.

**Informed Consent Statement:** Not applicable.

**Data Availability Statement:** The data presented in this study are available from the corresponding author upon reasonable request.

**Acknowledgments:** The authors thank the North Carolina Zoo's elephant animal care staff and the veterinary staff for their assistance in collecting and processing the samples.

**Conflicts of Interest:** The authors declare no conflict of interest.

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
