# Peer review of "Effect of Prolonged Serum Storage Time and Varied Temperatures on Biochemical Values in African Savanna Elephants (Loxodonta africana)"

_2673-5636, doi:10.3390/jzbg4010002_

Round 1
Reviewer 1 Report
The work allows the groups of care and management of elephants to have useful information for the correct clinical interpretation, preventive medicine and control of nutrient supply from blood biochemical values obtained in non-ideal conditions.

Author Response
- The main success of the work is associated with the honesty of the group of researchers in delimiting the scope of the work, as well as in indicating the reasons why the interpretation of its results must be adjusted to the conditions in which they could develop it.
- The results are consistent with the experience of other laboratories on the difficulty of obtaining viable results in studies referring to enzyme values from samples that were not deep-frozen after obtaining the serum.
- It is striking the high number of analytes that maintain, for a considerably high period, values similar to the zero sample, despite being subjected to conditions of high temperatures.
The selected statistical analyses allow easy recognition of the findings of the work. - Similar studies are suggested that consider the use of serum from other species, with a protocol similar to that used in the elephants of the study and similar habitat conditions for the participating individuals, in order to distinguish the possible influence of these species on the behavior of the hematic-biochemical components considered in this study.
- For all this, I believe that the work can be published and will be attractive to groups that daily carry out research and preventive medicine tasks in this and other species with situations similar to those presented in it.
Thank you for these comments.
Reviewer 2 Report
This research is undertaken to address the common problem faced by veterinarians working in the field. The results of the research will be highly useful for the wildlife veterinarians who hesitate to collect samples due to fearing the delay in analysing the samples that may hamper the results. I congratulate the authors for addressing the issue and undertaking the work. The rationale behind the research is appreciated. The manuscript is well written, experimental design is clear, supportive and sufficient to test the hypothesis. The citations are relevant and most of them are recent. I could not find any self citations as far i have searched. The listed tables and given data are clear, relevant, appropriate, easy to understand and interpret. Transportation and storage are the two major difficulties in clinical sample analysis for wildlife veterinarians. Though the manuscript addresses the issue about storage, impetus is not given to transportation. It will be worthy to mention the mode of transportation, distance between the zoo and the laboratory and the containers used. Moreover the authors need to mention whether the group separation viz, refrigeration, heated etc and aliquoting are made immediately after collection or it is done after reaching the laboratory. The authors can mention the feasibility of a correction factor that can be included for each specific type of storage to normalise the results for interpretation and diagnosis. What is the scope of arriving at the factor?. For example whether the reduction in BUN value is consistent and repeatable and if so whether the percentage of reduction is similar, that it can be used for arriving at the factor.
Author Response
Reviewer 2 Comments:
- This research is undertaken to address the common problem faced by veterinarians working in the field. The results of the research will be highly useful for the wildlife veterinarians who hesitate to collect samples due to fearing the delay in analyzing the samples that may hamper the results. I congratulate the authors for addressing the issue and undertaking the work. The rationale behind the research is appreciated. The manuscript is well written, experimental design is clear, supportive, and sufficient to test the hypothesis. The citations are relevant and most of them are recent. I could not find any self-citations as far I have searched. The listed tables and given data are clear, relevant, appropriate, easy to understand and interpret.
Thank you for these comments.
- Transportation and storage are the two major difficulties in clinical sample analysis for wildlife veterinarians. Though the manuscript addresses the issue about storage, impetus is not given to transportation. It will be worthy to mention the mode of transportation, distance between the zoo and the laboratory and the containers used.
Thank you for identifying this portion of our project which we did not define. The additional details have been added to the Materials and Methods section (lines 78-85, and 93-95)
- Moreover, the authors need to mention whether the group separation via, refrigeration, heated etc. and aliquoting are made immediately after collection or it is done after reaching the laboratory.
Group separation occurred immediately after collection when samples reached the veterinary hospital located on the grounds of the zoo (within 1 hour of blood collection). This information has been clarified in the Materials and Methods section. (lines 78-80)
- The authors can mention the feasibility of a correction factor that can be included for each specific type of storage to normalize the results for interpretation and diagnosis. What is the scope of arriving at the factor? For example, whether the reduction in BUN value is consistent and repeatable and if so whether the percentage of reduction is similar, that it can be used for arriving at the factor.
This is an intriguing possibility, and one that we considered. However, any correction formulae we provide would only be accurate for the specific storge temperatures evaluated, and we do not have sufficient data to predict analyte degradation at other temperatures. Larger studies with more subjects and many more storage temperature conditions (e.g. storage at every temperatures representing every 5°C between 2°C and 32°C), would provide a much stronger correction formula for each analyte. This is now briefly discussed in the Discussion section (lines 240-243)